# Where Graph Meets Heterogeneity: Multi-View Collaborative Graph Experts

**Zhihao Wu**[1]*, **Jinyu Cai**[2]*, **Yunhe Zhang**[3], **Jielong Lu**[1], **Zhaoliang Chen**[4], **Shuman Zhuang**[5], **Haishuai Wang**[1]†

[1]Zhejiang Key Laboratory of Accessible Perception and Intelligent Systems,
College of Computer Science and Technology, Zhejiang University
[2]Institute of Data Science, National University of Singapore
[3]Department of Computer and Information Science, University of Macau
[4]Department of Computer Science, Hong Kong Baptist University
[5]College of Computer and Data Science, Fuzhou University
`{zhihaowu1999,jinyucai1995,zhangyhannie}@gmail.com, jielonglu2022@163.com,`
`chenzl23@outlook.com, shumanzhuang@163.com, haishuai.wang@zju.edu.cn`

## Abstract

The convergence of graph learning and multi-view learning has propelled the emergence of multi-view graph neural networks (MGNNs), offering strong capabilities to address complex real-world data characterized by heterogeneous and interconnected information. While existing MGNNs exploit the potential of multi-view graphs, the inherent conflict persists between the two critical inductive biases of multi-view learning, consistency and complementarity. Consequently, the challenge of defining and resolving this tension in the new context of multi-view graphs remains largely underexplored. To bridge this gap, we propose **M**ulti-**v**iew **C**ollaborative **G**raph **E**xperts (MvCGE), a novel framework grounded in the Mixture-of-Experts (MoE) paradigm. MvCGE establishes architectural consistency through shared parameters while preserving complementarity via layer-wise collaborative graph experts, which are dynamically activated by a graph-aware routing mechanism that adapts to the structural nuances of each view. This dual-level design is further reinforced by two novel components: a load equilibrium loss to prevent expert collapse and ensure balanced specialization, and a graph discrepancy loss based on distributional divergence to enhance inter-view complementarity. Extensive experiments on diverse datasets demonstrate MvCGE's superiority.

## 1  Introduction

With the advancement of data collection and analysis capabilities, modern data is no longer treated as isolated or singular. Instead, entities are now understood to be deeply interconnected through complex and pervasive relationships, which are captured with increasing fidelity. Graphs, due to their universality, are widely used to represent these relationships as a powerful modeling tool. They excel at capturing complex dependencies and have become a cornerstone of machine learning, leading to advanced techniques for learning representations from structured data [Liang *et al.*, 2024; Cai *et al.*, 2024; Dai *et al.*, 2016]. Beyond inter-entity relationships, the 'non-isolation' of data is also manifested in the heterogeneous information sources, referred to as different views. For instance, in social networks, a user may be associated with various types of interactions (e.g., friendship ties vs. behavioral interactions); With multimedia systems, objects can be described using multiple

---

*Equal contribution.
†Corresponding author: Haishuai Wang.

39th Conference on Neural Information Processing Systems (NeurIPS 2025).

modalities (e.g., visual features and semantic tags). This diversity introduces unique challenges, giving rise to new paradigms like multi-view learning [Liang *et al.*, 2025; Zhang *et al.*, 2025; Chen *et al.*, 2023a; Xu *et al.*, 2015]. For such heterogeneous data, prior work in multi-view learning provides a foundational framework built upon two well-known key properties. Specifically, each view provides unique information that complements the others, which is the property known as complementarity. Since these views are rooted in the same underlying entities, there is an inherent agreement between them, referred to as consistency. These two distinct characteristics enable multi-view learning to outperform the single-view counterpart [Tao *et al.*, 2019]. But these characteristics may inherently conflict [Dong *et al.*, 2023]. Overemphasis on either distinctive or shared information will diminish the advantage of multi-view learning. Therefore, the effective promotion of a balance between these two properties has become a key focus.

Both the universal inter-relationships and increasing views further extend the heterogeneity of data from different perspectives. But interestingly, graph learning uses topology as a unifying structure, providing a common ground to anchor, align, and integrate diverse information. Researchers have found that graph-based multi-view learning demonstrates promising performance [Li *et al.*, 2025; Wan *et al.*, 2024]. For example, Zhang *et al.*; Chen *et al.* facilitated view alignment by constructing graphs, which significantly enhanced clustering; while Duan *et al.*; Huang *et al.* focused on semi-supervised learning, fully exploiting the graph to amplify supervisory signals. However, these methods remain traditional, lacking a deeper understanding of graph theory, and mainly build on the concept of similarity preservation, struggling to cope with complex data. There is widespread interest in empowering multi-view learning with advanced graph techniques in the era of deep learning.

Recently, Graph Neural Networks (GNNs) have rapidly developed and become a powerful technology for graph learning. Spectral GNNs [Defferrard *et al.*, 2016] effectively extend convolutions to non-Euclidean graph data; Graph Convolutional Network (GCN) [Kipf and Welling, 2017], established upon this, was then proposed and quickly became a phenomenal model. Due to the powerful capabilities of GNNs [Xu *et al.*, 2019] and the synergistic effect between multi-view learning and graphs, Multi-view GNNs (MGNNs) have become an effective paradigm. Distinct from prior works, MGNNs typically explicitly formulate heterogeneous data as multi-view graphs and utilize modern GNN architectures. Although the graph structure is a powerful instrument for harmonizing heterogene-

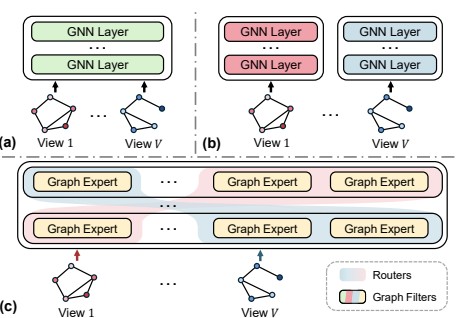

Figure 1: The illustration of MGNN paradigms based on (a) shared model, (b) multi-channel model, and (c) MvCGE.

ity, the classic dilemma between consistency and complementarity persists. The primary challenge shifts to *how to define and resolve this long-standing conflict within the new context of multi-view graphs.* Existing methods have made attempts, specifically, many MGNNs adopt a unified model that naturally allows for high collaboration between views through shared weights [Lin *et al.*, 2024; Chen *et al.*, 2023b; Cheng *et al.*, 2020; Xia *et al.*, 2022]. But this may lead to an overemphasis on consistency, thereby undermining complementarity. Also, in graph signal processing, a graph filter may fail to match the potentially heterogeneous multi-view graph structures. Another pipeline builds a multi-channel model [Chen *et al.*, 2024a; Lu *et al.*, 2024a; Wu *et al.*, 2023; Li *et al.*, 2020], where unique parameters are assigned to each view, ensuring complementarity. Because the interaction between views relies solely on post-fusion, this approach may sacrifice intrinsic inter-view interactions and lack architectural flexibility. The two existing pipelines are depicted in Figure 1. This raises a critical question: *How can we design a reasonable MGNN paradigm that achieves a balance between consistency and complementarity?*

In response to this challenge, we introduce a flexible and effective MGNN framework, termed **M**ulti-**v**iew **C**ollaborative **G**raph **E**xperts (MvCGE) in this paper. Figure 2 briefly illustrates the architecture of MvCGE. First, by reviewing existing MGNNs through the lens of graph spectral theory, we identify a critical limitation: shared graph filters cannot effectively handle the heterogeneous structural properties of multi-view graphs. Drawing inspiration from Mixture of Experts (MoE), we propose layer-wise collaborative graph experts as the foundational building block of MvCGE. This block enables distinct graph filters, or regarded as GNN layers, to collaboratively process multi-view

graphs, thus preserving view-specific information. Notably, the experts are shared across views, establishing an architectural foundation for consistency. Moreover, the number of graph experts is decoupled from the number of views, offering better flexibility. The incorporated graph-aware routing allows for the dynamic selection of suitable graph experts based on the structural characteristics of each view. To safeguard against model collapse and the consequent loss of complementary information, we introduce two novel mechanisms. The load equilibrium loss ensures a balanced allocation of training effort among the experts, avoiding dominance or over-proliferation of certain experts. The graph discrepancy loss, grounded in graph isomorphism, treats multi-view graph latent representations as discrete distributions and encourages experts to learn distinct representations, preventing collapse into the same latent space. The main contributions of this paper are:

- We thoroughly review existing MGNN pipelines and reveal a critical challenge, i.e., shared or view-specific models fail to flexibly adapt to multi-view graphs.

- We propose MvCGE, a novel MoE-inspired MGNN framework, effectively balancing requirements for consistency and complementarity in handling heterogeneous data.

- Two innovative mechanisms are introduced to address model collapse and enhance the protection of view-specific information from different aspects.

- Experiments on both single- and multi-view graph datasets demonstrate that MvCGE outperforms competitors, showcasing impressive performance and generalization capabilities.

## 2 Related Work

**Multi-view Learning** Multi-view learning [Liu *et al.*, 2023, 2022; Chen *et al.*, 2022a] leverages information from multiple perspectives to enhance performance, with consistency capturing shared information and complementarity harnessing unique contributions. Graph-based approaches, known for modeling arbitrarily distributed data, have gained prominence in this field [Sun *et al.*, 2024; Wan *et al.*, 2023; Li *et al.*, 2019]. Nie *et al.* proposed to automatically weight view-wise graphs in multi-view clustering. Li *et al.* utilized graphs to enhance incomplete multi-view learning. Wen *et al.* leveraged graphs to refine cross-view correspondences. However, these methods fail to fully exploit graphs' potential and lack integration with deep learning advancements.

**Mixture of Experts** The concept of MoE can be traced back to earlier works [Jacobs *et al.*, 1991; Jordan and Jacobs, 1994], designed for dynamically combining multiple specialized models to solve complex tasks. Modern advancements have adapted MoE to deep learning, making it powerful in vision and language models [Shazeer *et al.*, 2017; Lepikhin *et al.*, 2021; Fedus *et al.*, 2022; Fang *et al.*, 2022; Shen *et al.*, 2025; Miao *et al.*, 2025]. However, the potential of MoE in multi-view learning remains unexplored. Meanwhile, graph MoE is still in its infancy, with only a few researchers attempting to develop MoE for graph-based tasks [Ma *et al.*, 2024; Wang *et al.*, 2023; Hu *et al.*, 2022]. There remains a lack of investigation for multi-view graphs [Wang *et al.*, 2023].

**Graph Neural Networks** GNNs are a crucial tool for graph-based machine learning [Cai *et al.*, 2025; Wang *et al.*, 2025; Fang *et al.*, 2025b,a] and can be broadly divided into spectral and spatial approaches. Spectral GNNs, utilizing spectral graph theory to extend convolution operations to non-Euclidean space, once dominated the field. Kipf and Welling introduced the GCN, bridging the gap between spectral and spatial paradigms, and laying the foundation for modern GNNs. Building on these successes, MGNNs have emerged to tackle challenges in multi-view learning. Current MGNNs follow two main pipelines. One strategy constructs a unified model with shared parameters to process all views jointly, inherently ensuring consistency across views. To achieve this goal, Chen *et al.* proposed fusing multi-view graphs, while Lu *et al.* developed a graph generation approach. The other strategy employs multi-channel architectures, learning separate parameters for each view, as exemplified by [Lin *et al.*, 2024; Chen *et al.*, 2024b]. However, these two pipelines often lean toward two extremes: overemphasizing either consistency or complementarity. The former sacrifices view-specific information, while the latter may cause divergences and misalignments between views.

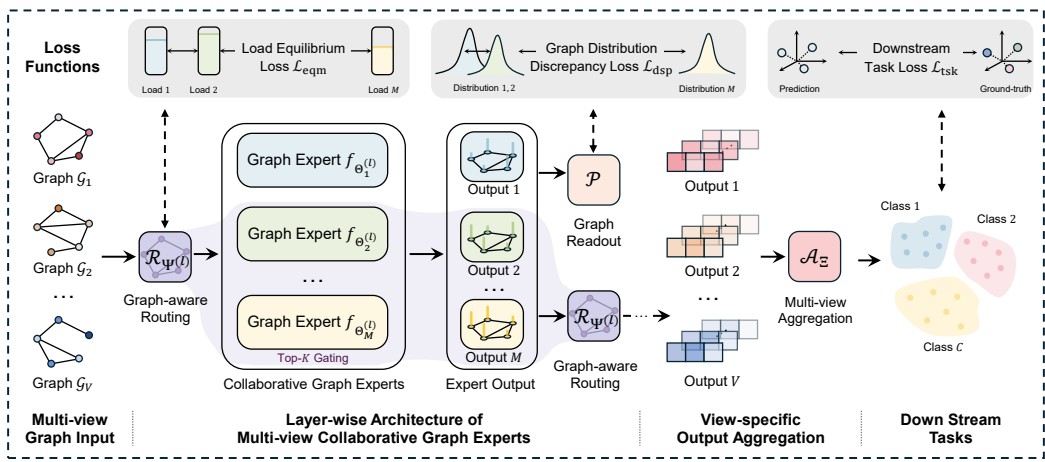

Figure 2: The illustration of a single layer of the proposed MvCGE.

## 3  Methodology

**Notation**   We define an attributed graph as $\mathcal{G} = (\mathcal{V}, \mathcal{E}, \mathbf{X})$, where $\mathcal{V}$ is the node set with cardinality $|\mathcal{V}| = N$, and $\mathcal{E}$ is the edge set. The graph's topology is represented by its adjacency matrix $\mathbf{A} \in \{0, 1\}^{N \times N}$. The node attributes are represented by a feature matrix $\mathbf{X} \in \mathbb{R}^{N \times D}$, where $D$ is the dimension of the node features. The degree matrix $\mathbf{D} \in \mathbb{R}^{N \times N}$ is a diagonal matrix where $\mathbf{D}_{ii} = \sum_j \mathbf{A}_{ij}$. The graph Laplacian is defined as $\mathbf{L} = \mathbf{D} - \mathbf{A}$ or $\mathbf{L} = \mathbf{I} - \mathbf{D}^{-\frac{1}{2}} \mathbf{A} \mathbf{D}^{-\frac{1}{2}}$ for its symmetric version. A multi-view graph describes a set of graphs $\{\mathcal{G}_v : v \in [V]\}$ where the node set $\mathcal{V}$ is associated with heterogeneous information. The $v$-th graph can be defined as $\mathcal{G}_v = (\mathcal{V}, \mathcal{E}_v, \mathbf{X})$ with corresponding adjacency matrix $\mathbf{A}_v$ and shared attribute $\mathbf{X}$.

**Motivation**   In this subsection, we will discuss existing paradigms from the lens of graph filters. To begin with, it is known that the Laplacian matrix can be decomposed as $\mathbf{L} = \mathbf{U} \mathbf{\Lambda} \mathbf{U}^\top$, which essentially induces a Fourier transform on the graph domain, where eigenvectors correspond to Fourier components and eigenvalues represent frequencies of the graph. Take $\mathbf{X}$ as the input signal, the general form of a graph convolutional layer can be written as

$$\hat{\mathbf{X}} = f_\Theta(\mathcal{G}, \mathbf{X}), \tag{1}$$

where $f_\Theta$ is a function parameterized with $\Theta$. For the early spectral GNNs [Bruna *et al.*, 2014], the function is specified as $h_\Theta(\mathbf{L})\mathbf{X}$, where $h_\Theta(\mathbf{L})$ is a parameterized graph filter. Due to the high computational burden, researchers then proposed to approximate the graph filter by a polynomial [Defferrard *et al.*, 2016], as

$$f_\Theta(\mathcal{G}, \mathbf{X}) = \sum_{t=0}^{T} \theta_t \mathbf{L}^t \mathbf{X}, \tag{2}$$

where $\Theta = \{\theta_t : t \in [T]\}$ are the parameters. [Defferrard *et al.*, 2016] achieved fast localized filters by using Chebyshev expansion, and the famous GCN [Kipf and Welling, 2017] further fixed $T = 2$ and simplified the convolution to bridge the spectral and spatial GNNs in a unified form. Therefore, without loss of generality, we describe the recursive formula of multi-layer modern GNNs:

$$\mathbf{H}^{(l+1)} = f_{\Theta^{(l)}}(\mathcal{G}, \mathbf{H}^{(l)}), \tag{3}$$

where $\mathbf{H}^{(l)} \in \mathbb{R}^{N \times D_l}$ is the input of the $l$-th layer, which is initialized by $\mathbf{H}^{(0)} = \mathbf{X}$, and $\Theta^{(l)}$ can be regarded as the extended multi-channel parameters of the graph filter. Training GNNs through optimizing the graph filter parameters $\Theta^{(l)}$ in each layer results in their powerful performance in graph-based machine learning tasks.

These GNNs have been rapidly applied to more complex and realistic multi-view graph scenarios. To ensure consistency across views, many approaches typically adopt a unified and shared model across all views. This strategy naturally enables highly collaborative learning among views by the weight

sharing, which leads to good consistency, but it often struggles to effectively model the multiplex graph structures inherent to diverse views. Considering any two views, we typically aim for the graph filter to produce results that exhibit consensus rather than significant disagreement. However, the effectiveness of a shared graph filter applied to two graph structures is significantly influenced by the spectral gap between these graphs. Discussions on the transferability or stability of graph filters are usually based on demanding assumptions [Ruiz *et al.*, 2020; Gama *et al.*, 2020; Levie *et al.*, 2021]. In other words, when the structures of the two graphs differ substantially, it becomes challenging to ensure that the graph filter achieves optimal performance across both views. An intuitive solution is to assign a separate graph filter to each view, allowing each view to learn parameters tailored to its own needs. This approach has been widely considered. However, while it takes into account more complementarities compared to a shared filter, it inevitably swings to the other extreme by sacrificing consistency in the architecture. Furthermore, the number of sub-modules is directly related to the number of views, which limits its flexibility. To address these limitations, we aim to engineer an MGNN framework that ensures scalable integration of inter-view dynamics, thereby better capturing and balancing the consistency and complementarity.

**Overall Framework**    To begin with, let us consider the input from the $v$-th view and define the following layer-wise formulation:

$$\mathbf{H}_v^{(l+1)} = \mathcal{F}_{\Phi^{(l)}}(\mathcal{G}_v, \mathbf{H}_v^{(l)}), \tag{4}$$

where $\mathbf{H}_v^{(l)}$ denotes the $v$-th graph embedding at the $l$-th layer with $\mathbf{H}_v^{(0)} = \mathbf{X}$, and the function $\mathcal{F}_{\Phi^{(l)}}$ with trainable parameter set $\Phi^{(l)}$ is assembled by multiple graph convolutions:

$$\mathcal{F}_{\Phi^{(l)}}(\mathcal{G}_v, \mathbf{H}_v^{(l)}) = \mathcal{R}_{\Psi^{(l)}}\big(\big\{f_{\Theta_m^{(l)}}(\mathcal{G}_v, \mathbf{H}_v^{(l)}) : m \in [M]\big\}\big), \tag{5}$$

where $\mathcal{R}_{\Psi^{(l)}}$ denotes the routing function parameterized by $\Psi^{(l)}$, and $\Phi^{(l)} = \{\Psi^{(l)}, \Theta_m^{(l)} : m \in [M]\}$. Building upon this, we introduce an ensemble of multiple graph filters, $\mathcal{F}_{\Phi^{(l)}}$, aiming to leverage graph filters equipped with distinct parameters to collaboratively process graph information from different views. Specifically, by designing the router $\mathcal{R}_{\Psi^{(l)}}$, appropriate combinations of graph filters are selected to handle potentially heterogeneous structures from various views. Subsequently, we consider processing a multi-view graph and constructing a multi-layer network. The proposed MvCGE framework calculates the final representation $\mathbf{H} \in \mathbb{R}^{N \times C}$ by

$$\mathbf{H} = \mathcal{A}_\Xi\big(\big\{\mathbf{H}_v = \mathcal{F}_\Phi(\mathcal{G}_v, \mathbf{X}) : v \in [V]\big\}\big), \tag{6}$$

where $\mathbf{H}_v \in \mathbb{R}^{N \times C}$ is the $v$-th learned final embedding, $\mathcal{A}_\Xi$ is a multi-view aggregation function with parameter set $\Xi$, and $\mathcal{F}_\Phi$ is defined by concatenating $L$ layer-wise functions:

$$\mathcal{F}_\Phi := \mathcal{F}_{\Phi^{(0)}} \circ \cdots \circ \mathcal{F}_{\Phi^{(l)}} \circ \cdots \circ \mathcal{F}_{\Phi^{(L-1)}}, \tag{7}$$

The entire function $\mathcal{F}_{\Phi^{(l)}}$ is expected to be shared across views, thereby facilitating inherent inter-view interactions. In our framework, $\mathcal{F}_\Phi$ is shared across views, and this characteristic ensures sufficient interaction between the graphs of different views at the architectural level. That is, all parameters $\Phi$ in the model are trained across the graphs of all views to capture consensus. Additionally, each layer $\mathcal{F}_{\Phi^{(l)}}$ of $\mathcal{F}_\Phi$ integrates multiple graph experts $f_{\Theta_m^{(l)}}$, which ensures that the model has sufficient capacity to adapt to the data of each view, thereby capturing their complementarities. Note that the number of graph experts $M$ is independent of the number of views $V$, which provides the model with additional flexibility. Therefore, MvCGE strikes a balance between the inherent requirements for consistency and complementarity in multi-view learning. After outlining the framework of MvCGE, the following subsection will discuss the details and specific implementation.

**Layer-wise Collaborative Graph Experts**    For the aforementioned design ideas, we implement our framework by drawing inspiration from the concept of MoE. Formally, given the input of the $v$-th view in the $l$-th layer $\mathbf{H}_v^{(l)}$, we specify that $\mathcal{F}_{\Phi^{(l)}}$ aggregate decisions from several graph experts by

$$[\mathbf{H}_v^{(l+1)}]_{i,:} = \sum_{m \in [M]} [\boldsymbol{\Gamma}_v^{(l)}]_{i,m} f_{\boldsymbol{\Theta}_m^{(l)}}(\mathcal{G}_v, \mathbf{H}_v^{(l)})_{i,:}, \tag{8}$$

where $\boldsymbol{\Gamma}_v^{(l)} \in \mathbb{R}^{N \times M}$ is the $v$-th routing matrix that contains the routing weights of all the graph filters in the $l$-th layer, which acts as graph experts in this framework. Through this layer-wise design,

each layer of MvCGE can be viewed as an independent MoE and is controlled by a specific router, allowing for more flexible selection and combination of graph experts.

Next, we specify the routing function $\mathcal{R}_{\Psi^{(l)}}$, whose role is to select the appropriate combinations of experts for each node in each view. We propose an improved graph-aware top-$k$ gating mechanism. Specifically, we denote the gating weight of the $i$-th sample in the $v$-th view to the $m$-th expert as

$$[\mathbf{\Gamma}_v^{(l)}]_{i,m} = \text{Softmax}\big(g_{\Psi^{(l)}}(\mathcal{G}_v, \mathbf{H}_v^{(l)})_{i,:}\big) \quad \text{s.t.} \ \big\|[\mathbf{\Gamma}_v^{(l)}]_{i,:}\big\|_0 = K, \ \forall i \in [N],$$

where the gating module $g_{\Psi^{(l)}}$ outputs an $M$-dimensional vector acting as the weights for integrating outputs from diverse experts. Note that this module has a constraint to ensure sparse gating. In multi-view graph learning scenarios, the design of $g_{\Psi^{(l)}}$ is critical, as it is key to preserving view specificity. It must both perceive the graph and view-level patterns.

$$g_{\Psi^{(l)}}(\mathcal{G}_v, \mathbf{H}_v^{(l)})_{i,:} = \text{Top}K\Big([\mathcal{P}(\{[\mathbf{H}_v^{(l)}]_{j,:}, j \in \mathcal{S}_i\})\|\mathcal{P}(\mathbf{H}_v^{(l)})]\mathbf{W}_{\text{gate}}^{(l)}\Big). \tag{9}$$

where $\mathcal{P}$ is a pooling function and $\mathcal{S}_i$ is the node set associated with a sampled local subgraph of node $i$. Therefore, $\mathcal{P}(\{[\mathbf{H}_v^{(l)}]_{i,j}, j \in \mathcal{S}_i\})$ and $\mathcal{P}(\mathbf{H}_v^{(l)})$ respectively encode the local graph structure and view-specific global information. $\mathbf{W}_{\text{gate}} \in \mathbb{R}^{2D \times M}$ is the trainable parameter matrix mapping the input to gating scores, respectively. We adopt Top-$K$ sparse gating: for each node $i$, only the Top-$K$ expert logits are kept while the rest are masked out (set to $-\infty$ before applying softmax). This yields $\|[\mathbf{\Gamma}_v^{(l)}]_{i,:}\|_0 = K$ with the remaining weights being exactly zero, such that only the experts with top-$K$ gating weight are selected.

**Acceleration Trick**  When processing large-scale graphs, graph convolution operations can become a major bottleneck due to the massive number of edges. The employment of multiple graph experts imposes additional computational overhead. Therefore, we consider further specializing the graph experts and propose an acceleration trick. We formalize the graph expert as

$$f_{\Theta_m^{(l)}}(\mathcal{G}_v, \mathbf{H}_v^{(l)}) = \zeta(\mathbf{P}_v \mathbf{H}_v^{(l)} \mathbf{\Theta}_m^{(l)}), \tag{10}$$

where $\zeta$ is a non-linear activation function, $\mathbf{P}_v$ is a filtering operator for the view $v$, and $\mathbf{\Theta}_m^{(l)} \in \mathbb{R}^{D_l \times D_{l+1}}$ is the trainable weight matrix. Here, we consider a class of simplified spectral GNNs where the filter parameters are treated as being encoded in the trainable weights $\mathbf{\Theta}_m^{(l)}$, thereby decoupling a parameter-free operator $\mathbf{P}_v$. For example, it can be the Chebyshev polynomial $\sum_{t=0}^{T} \mathrm{T}_t(\mathbf{L}_v)$ for ChebNet [Defferrard *et al.*, 2016], and $\mathbf{I} - \mathbf{L}_v$ for GCN [Kipf and Welling, 2017]. For this class of graph experts, we only need to compute the parameter-free graph filtering once per MvCGE layer. This result is then shared among different experts, which apply their distinct trainable weights for different graph convolutions. This considerably accelerates the entire framework.

**Regularization**  Although MvCGE ensures consistency through a flexible architecture, its training faces challenges that could lead to model collapse, thereby losing the complementary information between views.

*Load Equilibrium:* From the optimization perspective, due to the gating mechanism [Shazeer *et al.*, 2017], in some cases MvCGE may overly rely on a group of graph experts while preventing others from being trained. This imbalance is self-reinforcing, causing the selected experts to proliferate, while other experts become increasingly unlikely to be chosen. Ultimately, MvCGE will collapse into a model with only a few experts activated, losing the advantage of capturing view-specific information. To address this, we introduce a load equilibrium loss. For the $v$-th view, let

$$\rho_v^m = \frac{1}{N} \sum_i [\mathbf{\Gamma}_v^{(l)}]_{i,m}, \ \omega_v^m = \frac{1}{N} \sum_i \mathbb{I}\{[\mathbf{\Gamma}_v^{(l)}]_{i,m} > 0\}$$

be the average gating weight and load of expert $m$, we have

$$\mathcal{L}_{\text{eqm}} = \frac{1}{V} \sum_{v=1}^{V} \frac{1}{M} \sum_{m=1}^{M} \omega_v^m \cdot \rho_v^m \log \rho_v^m, \tag{11}$$

where we consider the intra-view and then the inter-view equilibrium. By minimizing $\mathcal{L}_{\text{eqm}}$, each graph experts are encouraged to have similar selection scores and frequency.

*Graph Discrepancy:* From the perspective of the latent space, even if load equilibrium is ensured and different graph experts are given equal training opportunities, they may collapse into similar latent spaces. Different graph experts are supposed to perform distinct filtering on multi-view graphs, rather than collapsing into a single graph filter. To address this challenge, we take a single-layer MvCGE as an example to elucidate the computational mechanism of the graph discrepancy loss. Given the $v$-th view graph $\mathcal{G}_v$, let $f_{\boldsymbol{\Theta}_m}$ and $f_{\boldsymbol{\Theta}_{m'}}$ denote any two distinct graph experts, the expert-specific embeddings are defined as:

$$[\mathbf{Z}_v^m]_{i,:} = [\boldsymbol{\Gamma}_v]_{i,m} f_{\boldsymbol{\Theta}_m}(\mathcal{G}_v, \mathbf{X})_{i,:}, \tag{12}$$

where $\boldsymbol{\Gamma}_v$ denotes the routing matrix and $\mathbf{Z}_v^m$ denotes the expert-specific embedding generated by expert $f_{\boldsymbol{\Theta}_m}$. Similarly, we obtain $\mathbf{Z}_v^{m'}$ for the other graph expert. For clarity, we omit the layer index in this single-layer illustration. Then the two embeddings are fed into the readout function, or called pooling function as

$$\mathbf{z}_v^m = \mathcal{P}(\mathbf{Z}_v^m). \tag{13}$$

Here, $\mathbf{z}_v^m$ can be interpreted as a global graph embedding that encapsulates the semantic and structural information captured by expert $f_{\boldsymbol{\Theta}_m}$. Recent studies have demonstrated that readout functions paired with GNNs can effectively perform graph isomorphism computations [Xu *et al.*, 2019]. Therefore, we combine the readout function with a layer of GNN to form a model $\mathcal{Q}$. Then for all $V$ views, these graph embeddings $\{\mathbf{z}_v^m : v \in [V]\}$ are treated as samples drawn from a distribution $\mathbb{P}_{\mathcal{Q}_m}$ induced by the function $f_{\boldsymbol{\Theta}_m}$ where $\mathcal{Q}_m = \mathcal{P} \circ f_{\boldsymbol{\Theta}_m}$.

After that, we leverage Maximum Mean Discrepancy (MMD) [Gretton *et al.*, 2012] to quantify the discrepancy between distributions from different graph experts:

$$\mathcal{M}_{\mathcal{H}}^2(\mathbb{P}_{\mathcal{Q}}, \mathbb{P}_{\mathcal{Q}'}) = \|\mu_{\mathcal{Q}_m} - \mu_{\mathcal{Q}_{m'}}\|_{\mathcal{H}}^2 = \mathbb{E}_{\mathbb{P}_{\mathcal{Q}}}[\kappa(\mathbf{z}_v^m, \mathbf{z}_{v'}^m)] + \mathbb{E}_{\mathbb{P}_{\mathcal{Q}'}}[\kappa(\mathbf{z}_v^{m'}, \mathbf{z}_{v'}^{m'})] - 2\mathbb{E}_{\mathbb{P}_{\mathcal{Q}}, \mathbb{P}_{\mathcal{Q}'}}[\kappa(\mathbf{z}_v^m, \mathbf{z}_{v'}^{m'})], \tag{14}$$

where $\kappa$ is a kernel function, e.g., Radial Basis Function kernel $\kappa(\mathbf{z}, \mathbf{z}') = \exp(\frac{\|\mathbf{z} - \mathbf{z}'\|^2}{2\sigma^2})$. Graph embeddings $\mathbf{z}$ and $\mathbf{z}'$ are sampled from distribution $\mathbb{P}_{\mathcal{Q}}$ and $\mathbb{P}_{\mathcal{Q}'}$ respectively. $\mu_{\mathbb{P}_{\mathcal{Q}}}$ and $\mu_{\mathbb{P}_{\mathcal{Q}'}}$ are the means of distributions $\mathbb{P}_{\mathcal{Q}_m}$ and $\mathbb{P}_{\mathcal{Q}_{m'}}$ in the reproducing kernel Hilbert space (RKHS) $\mathcal{H}$, respectively. Then we introduce the generalization of MMD

$$\mathcal{M}_{\mathcal{H}}^2(\mathbb{P}_{\mathcal{Q}}, \mathbb{P}_{\mathcal{Q}'}) = \sup\left\{\|\mu_{\mathcal{Q}} - \mu_{\mathcal{Q}'}\|_{\mathcal{H}}^2 : \kappa \in \mathcal{K}\right\}, \tag{15}$$

where $\mathcal{K}$ is a family of kernels. According to [Gretton *et al.*, 2012], we specifically estimate MMD by

$$\mathcal{M}_{\mathcal{H}}^2\left(\mathbb{P}_{\mathcal{Q}_m^{(l)}}, \mathbb{P}_{\mathcal{Q}_{m'}^{(l)}}\right) = \sup_{\kappa \in \mathcal{K}}\left[\frac{1}{V^2}\sum_v\sum_{v'}\kappa(\mathbf{z}_v^m, \mathbf{z}_{v'}^m) + \frac{1}{V^2}\sum_v\sum_{v'}\kappa(\mathbf{z}_v^{m'}, \mathbf{z}_{v'}^{m'}) - \frac{2}{V^2}\sum_v\sum_{v'}\kappa(\mathbf{z}_v^m, \mathbf{z}_{v'}^{m'})\right]. \tag{16}$$

Consequently, considering the multi-layer case, we propose the graph discrepancy loss:

$$\mathcal{L}_{\text{dsp}} = -\frac{1}{L}\sum_l \frac{1}{M(M-1)}\sum_{m<m'}\mathcal{M}_{\mathcal{H}}^2\left(\mathbb{P}_{\mathcal{Q}_m^{(l)}}, \mathbb{P}_{\mathcal{Q}_{m'}^{(l)}}\right), \tag{17}$$

where $\mathcal{Q}_m^{(l)} = \mathcal{P} \circ f_{\Theta_m^{(l)}}$ is a model combining a readout function and a graph expert. Utilizing the two losses together ensures that the graph experts receive similar opportunities and learn meaningful latent representations, thereby promoting MvCGE to capture complementary information.

**Training Strategy** The proposed MvCGE is trained using the following objective function

$$\mathcal{L}_{\text{total}} = \mathcal{L}_{\text{tsk}} + \alpha\mathcal{L}_{\text{eqm}} + \beta\mathcal{L}_{\text{dsp}}, \tag{18}$$

where $\mathcal{L}_{\text{tsk}}$ is the downstream task loss. The widely used Cross-Entropy loss is adopted for the semi-supervised classification task:

$$\mathcal{L}_{\text{tsk}} = -\sum_{i \in \Omega}\sum_j \mathbf{Y}_{i,j} \log \hat{\mathbf{Y}}_{i,j}, \tag{19}$$

where $\hat{\mathbf{Y}}, \mathbf{Y} \in \mathbb{R}^{|\Omega| \times C}$ are predicted and groud-truth labels, and $\Omega$ is the training set. Note that $\hat{\mathbf{Y}}_{i,:} = \text{Softmax}(\mathbf{H}_{i,:})$.

Table 1: Macro F1 and Micro F1 scores of all methods on multi-view datasets (mean value of ten runs), where the best and the second-best results are highlighted in **orange** and **blue**, respectively.

| Datasets | ACM-M | | DBLP | | IMDB | | YELP | | AMINER | |
|---|---|---|---|---|---|---|---|---|---|---|
| Metrics | MaF1 | MiF1 | MaF1 | MiF1 | MaF1 | MiF1 | MaF1 | MiF1 | MaF1 | MiF1 |
| GCN | 0.7860 | 0.7881 | 0.9012 | 0.9163 | 0.2429 | 0.5544 | 0.5196 | 0.6743 | 0.6842 | 0.8156 |
| DGI | 0.2235 | 0.3690 | 0.2432 | 0.3763 | 0.2626 | 0.5519 | 0.5034 | 0.6831 | 0.3449 | 0.6460 |
| HAN | 0.9147 | 0.9139 | 0.8927 | 0.9039 | 0.2389 | 0.5586 | 0.4829 | 0.4893 | 0.7234 | 0.8476 |
| DMGI | 0.8666 | 0.8681 | 0.6567 | 0.7106 | 0.3533 | 0.5726 | 0.5161 | 0.6985 | 0.3029 | 0.6546 |
| IGNN | 0.8290 | 0.8270 | 0.8681 | 0.8750 | 0.4531 | 0.5481 | 0.6449 | 0.7123 | 0.7453 | 0.8521 |
| MRGCN | 0.8758 | 0.8745 | 0.8949 | 0.9047 | 0.4517 | 0.4769 | 0.5435 | 0.7370 | 0.7335 | 0.8294 |
| SSDCM | 0.8765 | 0.8763 | 0.8942 | 0.8990 | 0.4940 | 0.5910 | 0.5270 | 0.7020 | 0.2620 | 0.5548 |
| MHGCN | 0.8887 | 0.8907 | **0.9300** | **0.9360** | **0.5154** | 0.6423 | 0.6085 | 0.7328 | 0.7500 | 0.8517 |
| AMOGCN | **0.9241** | **0.9238** | 0.9227 | 0.9280 | 0.5016 | **0.6506** | **0.6753** | 0.7241 | **0.7570** | **0.8338** |
| HMGE | 0.9080 | 0.9066 | 0.9156 | 0.9235 | 0.3293 | 0.5735 | 0.5857 | **0.7484** | 0.7371 | 0.8478 |
| MvCGE | **0.9348** | **0.9245** | **0.9256** | **0.9317** | **0.5157** | **0.6883** | **0.7776** | **0.8008** | **0.7897** | **0.8640** |

Table 2: Macro F1 and Micro F1 scores of all methods on single-view datasets (mean value of ten runs), where the best and the second-best results are highlighted in **orange** and **blue**, respectively.

| Datasets | ACM | | Citeseer | | CoraFull | | Flickr | | UAI | |
|---|---|---|---|---|---|---|---|---|---|---|
| Metrics | MaF1 | MiF1 | MaF1 | MiF1 | MaF1 | MiF1 | MaF1 | MiF1 | MaF1 | MiF1 |
| GCN | 0.8827 | 0.8842 | 0.6523 | 0.6909 | 0.5277 | 0.6279 | 0.5002 | 0.5104 | 0.4506 | 0.5851 |
| SGC | 0.8116 | 0.8087 | 0.6157 | 0.6642 | 0.5320 | 0.6292 | 0.4422 | 0.5101 | 0.4679 | 0.5652 |
| APPNP | 0.8837 | 0.8824 | 0.6510 | 0.6972 | 0.5236 | 0.6395 | 0.5124 | 0.5212 | 0.4655 | 0.6189 |
| JKNet | 0.8524 | 0.8545 | 0.6813 | 0.7292 | 0.5126 | 0.6268 | 0.5112 | 0.5431 | 0.4134 | 0.5623 |
| DAGNN | 0.8699 | 0.8917 | 0.6812 | 0.7290 | 0.5353 | 0.6481 | 0.6214 | 0.6065 | 0.4778 | 0.5927 |
| GCNII | 0.8978 | 0.8980 | 0.6712 | 0.7064 | 0.5828 | 0.6403 | 0.5904 | 0.5771 | **0.4869** | 0.6382 |
| GNNHF | **0.9052** | **0.9067** | 0.6760 | 0.7249 | **0.5846** | **0.6672** | 0.5867 | 0.6078 | 0.4495 | 0.5764 |
| AMGNN | 0.8996 | 0.8993 | **0.6873** | **0.7292** | 0.5664 | 0.6185 | **0.7580** | **0.7569** | 0.4787 | **0.6431** |
| HiDNet | 0.8958 | 0.8961 | 0.6661 | 0.7146 | 0.5304 | 0.6531 | 0.6200 | 0.6333 | 0.4568 | 0.6165 |
| AGNN | 0.8973 | 0.8974 | 0.6264 | 0.6721 | 0.5065 | 0.5930 | 0.5721 | 0.5844 | 0.4697 | 0.6206 |
| MvCGE | **0.9110** | **0.9114** | **0.6908** | **0.7356** | **0.5924** | **0.6693** | **0.7675** | **0.7790** | **0.4797** | **0.6591** |

# 4 Experiments

**Experimental Setting**   To evaluate the effectiveness of MvCGE, we conducted experiments on ten datasets, including five multi-view graph datasets (ACM, DBLP, IMDB, YELP, AMINER) and five typical single-view graph datasets (ACM, Citeseer, CoraFull, Flickr, UAI) plus one large-scale multi-view graph Freebase and one large-scale single-view graph OGBN-arXiv. To adapt MvCGE to single-view graphs, we generate a supplementary view for each single-view dataset based on the $k$-Nearest Neighbor algorithm, where we set $k = 10$. Note that, to distinguish between the different ACM datasets, we denote the natively multi-view graph ACM as ACM-M. For different types of datasets, we compared MvCGE with various competitors, including nine methods designed for multi-view datasets (DGI [Velickovic *et al.*, 2019], HAN [Wang *et al.*, 2019], DMGI [Park *et al.*, 2020], IGNN [Gu *et al.*, 2020], MRGCN [Huang *et al.*, 2020], SSDCM [Mitra *et al.*, 2021], MHGCN [Yu *et al.*, 2022], AMOGCN [Chen *et al.*, 2024b], HMGE [Abdous *et al.*, 2024]) and ten methods designed for single-view datasets (GCN [Kipf and Welling, 2017], SGC [Wu *et al.*, 2019], APPNP [Gasteiger *et al.*, 2018], JKNet [Xu *et al.*, 2018], DAGNN [Liu *et al.*, 2020], GCNII [Chen *et al.*, 2020], GNNHF [Zhu *et al.*, 2021], AMGNN [Zhu *et al.*, 2023], HiDNet [Li *et al.*, 2024], AGNN [Chen *et al.*, 2023c]).

**Performance**   In this subsection, we conduct comprehensive evaluations of MvCGE's performance in semi-supervised classification tasks across diverse datasets, empirically demonstrating its superior performance compared to state-of-the-art methods. Experimental setups and analyses on these datasets are as follows: 1) *Multi-view Graphs.* Here, the training ratio is set to 20% and the Macro F1 and Micro F1 scores are recorded in Table 1. The experimental results reveal that the proposed

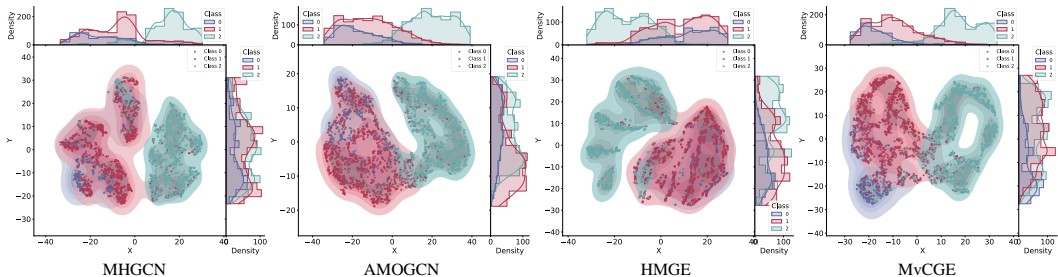

Figure 3: The visualization of representations learned by MHGCN, AMOGCN, HMGE, and MvCGE on the YELP dataset.

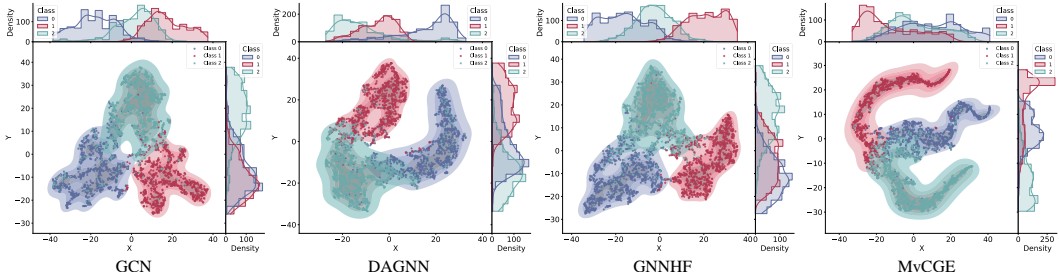

Figure 4: The visualization of representations learned by GCN, DAGNN, GNNHF, and MvCGE on the ACM dataset.

MvCGE outperforms baseline methods across most datasets. Notably, substantial performance gains are achieved on the Yelp dataset, primarily due to the inherent limitations of meta-path-based approaches in effectively modeling complex non-linear cross-view relationships. In contrast, guided by the principles of MoE, MvCGE effectively resolves the consistency-complementarity trade-off while preserving view-specific information in multi-view learning, as validated in our experiments. 2) *Single-view Graphs.* Following the commonly used semi-supervised node classification settings, we randomly select 20 samples per class for training, 500 samples for validation, and 1,000 samples for testing, with the detailed results presented in Table 2. The figure shows that MvCGE consistently outperforms all baseline methods and exhibits strong compatibility and adaptability to single-view datasets. Additionally, as depicted in Appendix, we also test MvCGE on the large-scale graph OGBN-arXiv and Freebase, showing its scalability.

**Visualization**   We employ t-SNE [Van der Maaten L, 2008] to visualize the learned representations of top-performing baselines on the YELP (Figures 3) and ACM datasets (Figures 4). Each node is represented as a point, color-coded by its class label, with contour lines indicating category density distributions. Complementary histograms along the axes reveal the embedding distributions across dimensions. The visualizations demonstrate that MvCGE achieves superior cluster separation with well-defined class boundaries and compact intra-class distributions. Analysis of the histogram distributions reveals that MvCGE achieves better-separated representations in the feature space.

**Sensitivity**   In this section, we assess the impact of varying the parameters $\alpha$ and $\beta$ on the performance of MvCGE, where $\alpha$ and $\beta$ balance the loss $\mathcal{L}_{\mathrm{eqm}}$ and $\mathcal{L}_{\mathrm{dsp}}$, as depicted in Figure 5 (a) and (b). It can be seen that model performance exhibits similar stable fluctuation patterns for both $\alpha$ and $\beta$ across ACM and YELP datasets, with the optimal performance achieved at $\alpha = 1e^{-1}$ or $1e^{-2}$ and $\beta = 1e^{-1}$ or $1e^{-3}$, demonstrating that the integrated two losses effectively capture complementary information and consistently enhance performance when appropriate parameter values are selected. Beyond this, we evaluated the model's sensitivity to the number of experts. Specifically, we varied the total number of experts and selected half of them, with the corresponding results presented in Figure 6 (a). In Figure 6 (b), we fixed the total number of experts at 10 while varying the number of selected experts. The findings indicate that choosing an optimal subset of experts, rather than an excessive or minimal number, enhances the model's performance.

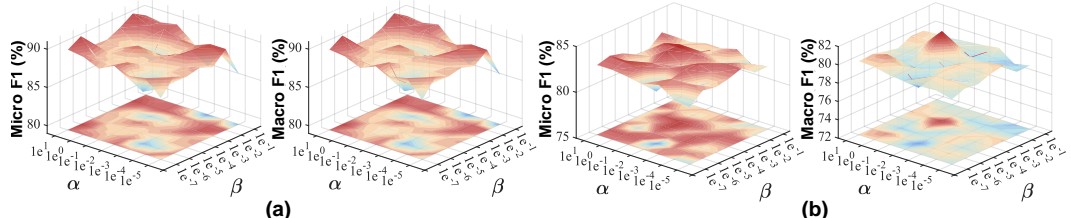

Figure 5: The classification performance of MvCGE w.r.t. hyperparameters $\alpha$ and $\beta$ on the (a) ACM dataset and (b) YELP dataset.

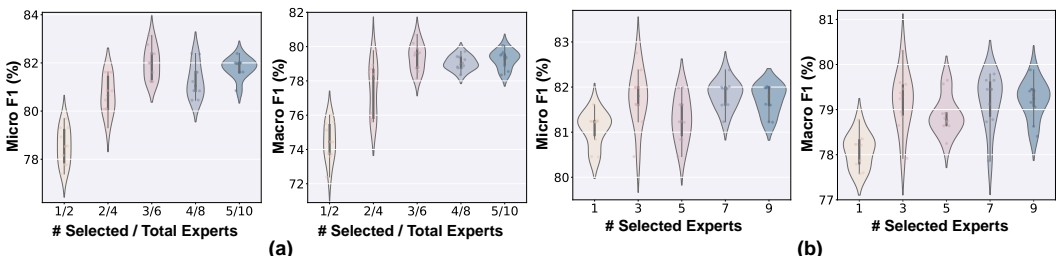

Figure 6: The classification performance of MvCGE w.r.t. (a) the number of selected and total graph experts and (b) the number of selected graph experts on the YELP dataset.

## 5 Conclusion

In this paper, we address the critical challenge of balancing consistency and complementarity in multi-view graph learning. Traditional multi-view GNNs often struggle to adapt to the heterogeneous structural properties of multi-view graphs due to rigid weight-sharing or isolated multi-channel designs. To overcome these limitations, we propose MvCGE, a novel framework inspired by Mixture of Experts, which introduces layer-wise collaborative graph experts to dynamically process multi-view graphs while preserving both shared and view-specific information. The integration of load equilibrium loss and graph discrepancy loss effectively mitigates model collapse and enhances the discriminative power of learned representations. Extensive experiments across diverse datasets demonstrate that MvCGE achieves state-of-the-art performance, showcasing its flexibility and robustness in handling complex multi-view graph data. One possible limitation is that we do not discuss the homophily and heterophily of graphs, although we adopt both types of graphs in our experiments.

## 6 Acknowledgements

This work was supported by the National Natural Science Foundation of China (Grant Nos. 62202422 and 62372408).

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
