# OpenReview forum: "Where Graph Meets Heterogeneity: Multi-View Collaborative Graph Experts"
_NeurIPS.cc/2025/Conference — NeurIPS 2025 poster_

### Official Review · Reviewer_7FN5 · 2025-06-29

**Clarity:** 3
**Significance:** 4
**Originality:** 4
**Rating:** 5
**Confidence:** 4

**Summary:**

This paper introduces Multi-view Collaborative Graph Mixture of Experts (MvCGE), a novel framework that unifies multi-view learning and graph neural networks. MvCGE employs a pool of layer-wise graph experts that are dynamically routed to each view via a graph-aware top-K gating mechanism. Consistency is enforced by sharing expert parameters across views, while complementarity is preserved through two regularizers. Extensive experiments on five multi-view and six single-view benchmarks demonstrate consistent performance gains over strong baselines.

**Questions:**

Does the combination of multiple GNNs and multiple graphs result in a high computational burden? If so, does the proposed approach include any improvement for scalability?

**Ethical Concerns:**

["NO or VERY MINOR ethics concerns only"]

**Final Justification:**

Most of my concerns have been addressed, and therefore I decide to maintain my score.

**Limitations:**

Yes. The authors note the omission of homophily/heterophily discussion. I recommend the authors to include a discussion (or empirical analysis) of how their method performs under varying graph homophily levels.

**Paper Formatting Concerns:**

No such issues.

**Quality:**

3

**Strengths And Weaknesses:**

Strengths:

1. The work is clearly motivated by the trade-off between consistency and complementarity in multi-view GNNs.
2. The proposed framework seems to be novel and bridge the research gap in multi-graph learning.
3. Most of the descriptions and formalizations in the methods are clear, and it is convinced that the proposed MoE-based framework can be a good solution.
4. Evaluation across ten datasets (including large-scale Freebase and arXiv in the appendix) with both multi-view and single-view settings demonstrates broad applicability.

Weaknesses:

1. As per my understanding, introducing multiple GNNs on multiple graphs will increase the runtime and memory footprint by a significant amount compared to the baselines. However, the authors claim that the proposed method is efficient and does not impose a significant computational burden. The authors should clarify this issue.
2. The model requires tuning on several hyperparameters $\alpha$, $\beta$, total experts $M$, and top-$k$. although explored their impact, the need to tune four interdependent hyperparameters may hinder practical adoption.
3. Some equations (e.g., the generalized MMD in Eq. 16 vs. 17) and definitions (routing mechanism) could be streamlined in the main paper and elaborated in the appendix, or readers may struggle with these.

---

> ### Author Rebuttal · Authors · 2025-07-31
>
> **W1/Q1. Complexity concerns**
>
> **Response:** Our solution introduces a minor additional overhead but does not lead to a considerable increase in overall complexity [1, 2]. MoE has been widely adopted in LLMs precisely because it allows for the expansion of the model capacity while maintaining inference efficiency through sparse expert activates [3, 4]. In this work, we provide a detailed theoretical complexity analysis in Appendix B. When employing $ E $ experts, the computational complexity of the gating mechanism within a single MvCGE layer is $ \mathcal{O}(NDE+|\mathcal{E}|D) $, while that of each hierarchical graph expert (e.g., GCN) is $ \mathcal{O}(ND^2+|\mathcal{E}|D) $. In practice, $ E $ is typically not much larger than the $ D $, so it will not be a bottleneck. The most time-consuming operation remains the graph convolution ($ \mathcal{O}(|\mathcal{E}|D) $). It is the gating mechanism that allows each sample to activate only a small subset of $K$ experts, even with dozens of graph experts. In addition, for spectral GNNs such as GCN, we can decouple the filter computation from the learnable parameters [5, 6], substantially reducing the number of filtering operations required. Empirically, Appendix A reports the results of MvCGE on two large-scale graphs, and here we further provide efficiency comparisons on three datasets to offer a clearer perspective:
>
> | Method | ACM (Time) | ACM (Memory) | Citeseer (Time) | Citeseer (Memory) | Flickr (Time) | Flickr (Memory) | UAI (Time) | UAI (Memory) |
> | --- | --- | --- | --- | --- | --- | --- | --- | --- |
> | **GCN** | 1.82s | 24.49 MB | 1.82s | 51.04 MB | 3.01s | 369.76 MB | 1.96s | 65.28 MB |
> | **MvCGE (3 experts, top 1)** | 3.60s | 25.41 MB | 3.33s | 52.21 MB | 4.91s | 374.16 MB | 3.52s | 67.00 MB |
> | **MvCGE (6 experts, top 2)** | 4.95s | 25.57 MB | 5.07s | 52.39 MB | 7.03s | 374.54 MB | 5.46s | 67.20 MB |
>
> **Response:** We evaluated MvCGE under two settings: (1) 3 experts with each sample selecting the top-1 expert, and (2) 6 experts with each sample selecting the top-2 experts. The experiments were conducted on single-view graphs. Note that GCN processes only a single view, whereas MvCGE processes two views simultaneously. Nevertheless, MvCGE did not introduce significant computational overhead, which is consistent with our theoretical analysis and other empirical results.
>
> **W2. Multiple hyperparameters tuning**
>
> **Response:** Although the proposed model does require tuning of four hyperparameters, in practice, we fixed the number of experts to 6 and the number of selected experts to 2, and still achieved strong performance across all datasets. Furthermore, as demonstrated in the experiments shown in Fig. 6, it is not necessary to finely tune these two hyperparameters within a reasonable range.
>
> **W3. Regarding equations and definitions**
>
> **Response:** We apologize for any inconvenience caused by the complex notation and formulas. We will carefully consider your suggestions and systematically adjust the organization and presentation of the text to facilitate readers’ understanding of our method.
>
> ```
> [1] Towards understanding the mixture-of-experts layer in deep learning. NeurIPS 2023.
> [2] Outrageously large neural networks: The sparsely-gated mixture-of-experts layer. ICLR 2017.
> [3] Switch transformers: Scaling to trillion parameter models with simple and efficient sparsity. JMLR 2022.
> [4] Scaling vision with sparse mixture of experts. NeurIPS 2021.
> [5] Semi-Supervised Classification with Graph Convolutional Networks. ICLR 2017.
> [6] Convolutional neural networks on graphs with chebyshev approximation, revisited. NeurIPS 2022.
> ```

---

### Official Review · Reviewer_zXFJ · 2025-06-29

**Clarity:** 3
**Significance:** 4
**Originality:** 4
**Rating:** 5
**Confidence:** 5

**Summary:**

This paper proposes a MoE-inspired framework to harmonize consistency and complementarity in multi-view graph learning. By employing shared collaborative graph experts and a graph-aware router, the model dynamically selects experts per view while maintaining architectural consistency. Two regularization strategies, load balancing and distributional divergence, prevent expert dominance and enhance view-specific representations. Experiments show state-of-the-art performance across 10 datasets.

**Questions:**

- How does homophily/heterophily interact with the proposed model?
- What do proposition 1 and corollary 1 suggest?
- See weaknesses for some questions.

**Ethical Concerns:**

["NO or VERY MINOR ethics concerns only"]

**Final Justification:**

This work presents both novelty and clear presentation, and it addresses a practical and important problem. This work is expected to have a positive impact on the graph learning community. The main concerns were related to the explanation of certain issues, most of which have been addressed in the authors' response. Therefore, I maintain the high score.

**Limitations:**

Yes the authors discuss limitations in the conclusion.

**Quality:**

4

**Strengths And Weaknesses:**

## Strength

- The paper is well-motivated and well-written. The authors do identify a critical challenge in multi-view graph learning and the design limitations of existing multi-view graph neural networks. The design principles of this work could inspire MoE applications in other graph tasks.
- One principal strength of this work is the proposed architecture decouples expert count from view count, which achieves dynamic adaptation to multi-view graphs and without composing much computational burden.
- Providing executable source code is encouraged, I run the code and can reproduce the reported results. BTW, the code is well-organized and easy to read.

## Weakness

- As the author mentioned in the conclusion, one limitation of this paper is the absence of a discussion on the interaction between homophily/heterophily and the proposed MvCGE. Addressing this issue may provide more insights into this work.
- I note that the authors provide some theoretical analysis in Appendix C, can the authors clarify the necessity of Proposition 1 and Corollary 1 and how they demonstrate the validity of the proposed approach?
- The discussion of graph discrepancy appears to lack sufficient definitional clarity. Regarding the distributions $\mathbb{P}\_{Q}$ and $\mathbb{P}\_{Q’}$, what exact probability measures do $\mathbb{P}\_{Q}$ and $\mathbb{P}\_{Q’}$ represent in the proposed framework? How to sample instances from these distributions to compute the MMD-based graph discrepancy $M_{H}(\mathbb{P}\_{Q}, \mathbb{P}\_{Q’})$?

---

> ### Author Rebuttal · Authors · 2025-07-31
>
> We sincerely thank the reviewer for their careful reading and recognition of our work. Below, we provide point-by-point responses to address your concerns.
>
> **W1/Q1. Discussion on homophily/heterophily**
>
> **Response:** In this work, we did not explicitly discuss the issue of homophily or heterophily. However, spectral GNNs are closely related to these properties [1, 2]. In homophilic graphs, neighboring nodes tend to share similar attributes or labels, implying that the energy is concentrated in the low-frequency components of the Laplacian spectrum [3, 4]. Conversely, heterophilic graphs exhibit substantial variations between nodes, leading to more high-frequency components. In multi-view graphs, despite sharing the same node set, different views may exhibit significant differences in homophily and heterophily, resulting in distinct spectral distributions. This heterogeneity renders single graph convolutions/filters inadequate for unified processing, which aligns with our theoretical motivation. For instance, the YELP dataset demonstrates substantial inter-view divergence, which explains why MvCGE achieves notably superior performance on it. We will strengthen the discussion of these aspects in the Method and Experiment sections.
>
> **W2/Q2. Further discussions on theoretical analysis**
>
> **Response:** Proposition 1 and Corollary 1 primarily serve as the theoretical motivation to expose the inherent limitations of shared graph filters in multi-view graph learning. These insights motivate us to explore collaboration graph filter and propose our solution: By designing a team of specialized graph filters under the MoE framework, MvCGE enables: (1) View-Specific Filter Selection: Each view dynamically activates the most suitable filter combination through graph-aware routing; (2)Collaborative Spectral Processing: Distinct filters specialize in handling complementary frequency bands across views. Therefore, the total error can be decomposed and now depends on the optimal combination that can be chosen for each view from all graph experts, circumventing the error growth bound in Proposition 1. We will reconsider the placement and presentation of these theoretical analyses in future versions.
>
> **W3. Graph discrepancy loss clarification**
>
> **Response:** In MvCGE, each graph expert is assigned a set of samples and generates node representations for them, and we expect these graph experts to learn different latent spaces during the training process, and thus need to measure the distribution differences between the learned representations among the graph experts. We build on research related to graph isomorphism computation [5] by introducing the readout function combined with graph experts to obtain graph embeddings, which are essentially global representations that encode graph structure. And the distribution of graph embeddings is used to measure the difference between graph experts. Therefore, we first perform a readout of the representations learned by each graph expert for a certain view by $\mathbf{z} = \mathcal{P}(\mathbf{Z})$.
> Subsequently, using $Q$ to represent the combination of the readout function and the graph expert, we treat the set of all these embeddings (across different views) as samples from the distribution $\mathbb{P}\_{Q}$. Hence, $\mathbb{P}\_{Q}$ is the empirical distribution over the graph-level embeddings ${\mathbf{z}}$. Whereas different graph experts have different distributions, denoting as $\mathbb{P}\_{Q}$ and $\mathbb{P}\_{Q'}$.
> Finally, we measure how different the distributions are by using the powerful Maximum Mean Discrepancy (MMD) with an RBF kernel. For a certain view, we compute the node embeddings via graph expert $m$, apply the readout function $\mathcal{P}$, and obtain a single embedding $\mathbf{z}$. Doing this for each view in the dataset yields a set of $\mathbf{z}$, which we treat as samples from $\mathbb{P}\_{Q}$, and $\mathbb{P}\_{Q}$ is simply the empirical distribution of these embeddings.
>
> ```
> [1] Graph neural networks for graphs with heterophily: A survey. arXiv preprint arXiv:2202.07082 (2022).
> [2] Addressing heterophily in graph anomaly detection: A perspective of graph spectrum. WWW 2023.
> [3] Revisiting heterophily for graph neural networks. NeurIPS 2022.
> [4] Beyond low-frequency information in graph convolutional networks. AAAI 2021.
> [5] How powerful are graph neural networks? ICLR 2018.
> ```

---

> > ### Comment · Reviewer_zXFJ · 2025-08-06
> >
> > Thank you for the response, which has addressed most of my concerns. Therefore, I will maintain my positive score.

---

> > > ### Author Response · Authors · 2025-08-08
> > >
> > > We sincerely appreciate your constructive comments and will thoroughly revise our manuscript according to the suggestions and findings discussed.

---

### Official Review · Reviewer_HqvP · 2025-06-30

**Clarity:** 2
**Significance:** 3
**Originality:** 3
**Rating:** 5
**Confidence:** 4

**Summary:**

This paper addresses the challenge of balancing consistency and complementarity in multi-view graph neural networks (MvGNNs) by introducing a Mixture-of-Experts–inspired framework called Multi-view Collaborative Graph Mixture of Experts (MvCGE). At each layer, a set of shared “graph experts” (i.e., distinct GNN filters) is dynamically routed per view via a graph-aware top-k gating mechanism. To prevent expert collapse and encourage diversity, the authors also propose two losses. Experiments on five multi-view and six single-view benchmarks demonstrate that MvCGE consistently outperforms baselines.

**Questions:**

Q1. How exactly is the routing function calculated? Equations 8, 10 and 11 are a bit confusing, can you provide a step-by-step description or a table of algorithm for better clarity?

Q2. In Equation 13, why does the superscript $(l)$ disappear and why is it inconsistent with Equation 8?

**Ethical Concerns:**

["NO or VERY MINOR ethics concerns only"]

**Final Justification:**

The authors have addressed my concerns regarding reproducibility and readability. The paper is novel and technically sound. I am raising my score and recommend acceptance.

**Limitations:**

Yes.

**Quality:**

3

**Strengths And Weaknesses:**

**Strengths**

S1. Novel and well-motivated. The framework innovatively addresses the key challenge of balancing consistency and complementarity in multi-view learning by combining MoE with GNN, has a practical impact and makes sense for the community.

S2. It is interesting that the work designs the graph discrepancy and the graph-aware router, which are capable of regularizing the experts and learning awareness of graph structures.

S3. The evaluation of the proposed method is comprehensive, experiments include performance comparisons over both single-view and multi-view graphs, sensitivity analyses, and visualization.

**Weaknesses**

W1. Reproducibility is doubtful. The authors have included only an appendix in the supplemental material and have not provided any source code or any form of pseudo-code for reproduction. This may affect the credibility of the experimental results.

W2. The Methodology section is hard to follow. The complex notations made me spend more time reading about the methodology, especially for the collaborative experts and graph discrepancy regularization. I list the questions in the following part.

W3. Lack of some literature. In related work, the authors only introduce a few MoE-GNN papers, and some recent works are not discussed or compared as baselines, leaving unclear how this work positions itself in the literature.

W4. Limited theoretical analysis. Beyond intuitive motivation, the paper lacks a formal or theoretical analysis of the proposed method.

---

> ### Author Rebuttal · Authors · 2025-07-31
>
> We appreciate your thorough review and insightful comments. Below we address every weakness (W) and question (Q) in detail, and we explain the concrete additions that will appear in our revised manuscript.
>
> **W1. Reproducibility**
>
> **Response:** To facilitate online access without the need to download additional files, we did not include the source code directly in the supplementary materials. Instead, we have provided an anonymous link for convenient viewing. We would like to clarify that the anonymous project link to the source code, as well as the implementation platform details, are both ***already included in Appendix A.2***. We believe these materials sufficiently demonstrate the effectiveness of the proposed MvCGE and ensure the reproducibility of the experiments presented in this paper.
>
> **W2/Q1. Details about the method**
>
> **Response:** We apologize for any inconvenience this may have caused. To ensure precise expression, we adopted a relatively complex notation system. In future versions, we will adjust the organization of the main text and appendices by simplifying the notation as much as possible in the main body, moving certain technical details to the appendix, and providing comprehensive explanations for each symbol. This will help balance readability and accuracy. Here we provide a step-by-step description of the routing function:
>
> | Step                          | Formula / code                                                                                               | Explanation                                               |
> | ----------------------------- | ------------------------------------------------------------------------------------------------------------ | --------------------------------------------------------- |
> | 1. **Aggregation** | $u_i = [P_v H^{(l)}_v]_i$                                                                                    | Graph convolution of node *i* under view *v*.             |
> | 2. **Raw scores**             | $s_i = u_i W_{\text{gate}}$                                                                                  | Linear map to E-dimensional gate scores.           |
> | 3. **Top-K sparsification**          | $s_i \leftarrow s_i \cdot \mathbf 1_{\{s_i \ge \delta\}}$  ($\delta = K$-th largest value)                                | Keeps only the $K$ strongest expert. |
> | 4. **Softmax**                | $[\Gamma^{(l)}\_v]\_{i,:} = \text{softmax}(s\_i)$                                           | Normalises surviving scores.                              |
> | 5. **Expert fusion**          | $h\^{(l+1)}\_{v,i} = \sum\_m [\Gamma^{(l)}\_v]_{i,m} \, f^{(l)}\_{\Theta\_m}(G\_v,H^{(l)}\_v)_i$   | Weighted sum yields node output.                          |
>
> **W3. Lack of GNN-MoE-based literature**
>
> **Response:** As discussed in the related work section, the MoE [1] architecture has been widely adopted in many modern large language models (LLMs) [2], yet its potential in graph learning remains largely underexplored. A few recent studies have proposed MoE-based methods for graph learning, but these primarily focus on single-view graphs [3, 4, 5]. In contrast, the MoE architecture is inherently well-suited for modeling complex multi-view graphs, which represents a clear research gap. Consequently, we were unable to identify directly comparable baselines in the literature. In this work, we adopt the classical single-view GMoE [3] as a baseline for comparison. We conduct the comparison on both single- and multi-view datasets:
>
> | Method   | ACM MaF1     | ACM MiF1     | DBLP MaF1    | DBLP MiF1    |
> |----------|--------------|--------------|--------------|--------------|
> | GCN      | 0.7860       | 0.7881       | 0.9012       | 0.9163       |
> | DGI      | 0.2235       | 0.3690       | 0.2432       | 0.3763       |
> | HAN      | 0.9147       | 0.9139       | 0.8927       | 0.9039       |
> | DMGI     | 0.8666       | 0.8681       | 0.6567       | 0.7106       |
> | IGNN     | 0.8290       | 0.8270       | 0.8681       | 0.8750       |
> | MRGCN   | 0.8758       | 0.8745       | 0.9007       | 0.9047       |
> | SSDCM    | 0.8765       | 0.8763       | 0.8942       | 0.8990       |
> | MHGCN    | 0.8887       | 0.8907       | 0.9300  | 0.9360   |
> | AMOGCN   | 0.9241  | 0.9238   | 0.9227       | 0.9280       |
> | HMGE     | 0.9080       | 0.9066       | 0.9156       | 0.9235       |
> | GMoE     |  0.8816  | 0.8808  | 0.8736  | 0.8858  |
> | MvCGE    | 0.9348   | 0.9245   | 0.9256   | 0.9317   |
>
> | Method   | ACM MaF1   | ACM MiF1   | Citeseer MaF1 | Citeseer MiF1 |
> |----------|------------|------------|---------------|---------------|
> | GCN      | 0.8827     | 0.8842     | 0.6523        | 0.6909        |
> | SGC      | 0.8116     | 0.8087     | 0.6157        | 0.6642        |
> | APPNP    | 0.8837     | 0.8824     | 0.6510        | 0.6972        |
> | JKNet    | 0.8524     | 0.8545     | 0.6813        | 0.7292        |
> | DAGNN    | 0.8699     | 0.8917     | 0.6812        | 0.7290        |
> | GCNIi    | 0.8978     | 0.8980     | 0.6712        | 0.7064        |
> | GNNHF    | 0.9052     | 0.9067     | 0.6760        | 0.7249        |
> | AMGNN    | 0.8996     | 0.8993     | 0.6873        | 0.7292        |
> | HiDNet   | 0.8958     | 0.8961     | 0.6661        | 0.7146        |
> | AGNN     | 0.8973     | 0.8974     | 0.6264        | 0.6721        |
> | GMoE     | 0.8964  | 0.8950 | 0.6789  | 0.7107  |
> | MvCGE    | 0.9110 | 0.9114 | 0.6908    | 0.7356    |
>
> As shown by the experimental results, GMoE demonstrates competitive performance on both types of datasets. However, it consistently falls short of MvCGE, further indicating that while the MoE architecture holds significant potential in multi-view graph scenarios, existing frameworks are insufficient to fully exploit this advantage.
>
> **W4. Limited theoretical analysis**
>
> **Response:** Although we do not provide a direct theoretical analysis of the proposed method, we do offer theoretical justification for its rationality. Specifically, a formal lower-bound analysis of single-filter transferability across views is presented in Appendix C. This analysis theoretically demonstrates that a single graph filter cannot adapt to multiple graph structures with distinct frequency characteristics, whereas a combination of multiple filters does not suffer from this limitation. Building upon this insight, we considered the balance between consistency and complementarity, which motivated the design of the MvCGE framework.
>
> **Q2. Regarding Equation (13)**
>
> **Response:** The section containing Equation (13) introduces the graph discrepancy loss, which measures the distributional differences between any pair of graph experts within each layer of MvCGE. Therefore, for the sake of clarity and conciseness, we omit the superscript $(l)$ and use $f_{\Theta_m}$ and $f_{\Theta_{m'}}$ to denote any two experts within the same layer when describing this loss.
>
> ```
> [1] Outrageously large neural networks: The sparsely-gated mixture-of-experts layer. ICLR 2017.
> [2] Deepseek-v3 technical report. arXiv preprint arXiv:2412.19437 (2024).
> [3] Graph mixture of experts: Learning on large-scale graphs with explicit diversity modeling. NeurIPS 2023.
> [4] Mixture of experts for node classification. ICMR 2025.
> [5] Fair graph representation learning via diverse mixture-of-experts. WWW 2023.
> ```

---

> > ### Comment · Reviewer_HqvP · 2025-08-07
> >
> > I appreciate the authors' rebuttal, which has clarified my misunderstandings regarding reproducibility. The improvements to the readability of the notations and the graph MoE baseline should be included into the camera ready version. In addition, I have one more question: Is the proposed method applicable to multi-modality graphs, and would any modifications needed? Overall, after reading the rebuttal, I find the paper to be novel and technically sound. I will raise my score and recommend acceptance, but the authors should improve their manusript based on the above discussion.

---

> > > ### Author Response · Authors · 2025-08-08
> > >
> > > Thank you for your encouraging feedback and for raising the critical question of how our method can handle multi-modal graphs. We appreciate your careful reading and will incorporate every suggestion into the revised manuscript. Regarding the extension to multi-modal data, we'd like to clarify that MvCGE was originally designed for graphs that share node features but differ in structure. However, by some modifications, the same framework can be seamlessly extended to multi-modal benchmarks [1]. Possible revisions include adding modality encoders, modality-aware routing and experts, and cross-modal regularization terms. We will consider more details in our revision.
> > >
> > > [1] Mosaic of Modalities: A Comprehensive Benchmark for Multimodal Graph Learning. CVPR 2025.

---

### Official Review · Reviewer_DaD6 · 2025-07-01

**Clarity:** 2
**Significance:** 2
**Originality:** 2
**Rating:** 4
**Confidence:** 4

**Summary:**

This paper identifies that existing multi-view GNNs either share parameters across views (over-emphasizing consistency but losing view-specific nuance) or allocate separate channels per view (preserving complementarity but sacrificing inter-view synergy). They propose MvCGE, which at each layer maintains a shared pool of graph-convolution experts and uses a graph-aware top-K routing to dynamically assign each view’s nodes to a subset of experts. A load-equilibrium loss ensures no expert is neglected, and a graph-discrepancy loss (via MMD) enforces that different experts learn distinct representations. Experiments on five multi-view and five single-view citation/dataset benchmarks show MvCGE consistently outperforms strong baselines in Macro-F1 and Micro-F1.

**Questions:**

1.  Scalability: How does inference time grow with graph size and expert count?
2.  How to define the view of multi-view graphs, is there any different designs of different experts ? Would different experts receive graph inputs only from that views, are different combinationn of designs have impacts towards final results ?
3.  Expert specialization: Have you observed that particular experts consistently specialize on certain views or substructures? Can these specializations be interpreted?
4.   Robustness to view noise: If one view is very noisy or missing, how does the routing mechanism compensate?
5.    Loss weighting: How sensitive is performance to the choice of α (load-equilibrium) and β (discrepancy)? Is there a principled way to set them?
6.    Generalization to other tasks: Can the same MvCGE framework be applied to link prediction or graph regression problems? What changes would be needed?

**Ethical Concerns:**

["NO or VERY MINOR ethics concerns only"]

**Final Justification:**

Thanks for the detailed reply. It resolve most of my concerns. I would like to raise my score accordingly.

**Limitations:**

yes

**Quality:**

2

**Strengths And Weaknesses:**

**Strengths**
- MOE for  multi-view GNN learning
By decoupling expert count from view count and dynamically routing, MvCGE achieves both architectural consistency (shared experts) and view complementarity (expert specialization) in a unified framework.
-   Novel Regularizers
The load-equilibrium loss prevents expert collapse (ensuring each expert is used), and the graph-discrepancy loss leverages MMD to keep expert representations distinct.
-   Strong Empirical Gains
On five multi-view graphs (e.g., ACM-M, YELP), MvCGE achieves good performance compared to other baselines, and it also improves over state-of-the-art single-view GNNs on standard benchmarks.

**Weaknesses**
- The Ldsp loss:
 This loss is to minimize the discrepancy between distributions from different graph experts. THIS DESIGN is actually against the orginal design to perform distinct filtering. Thus, this loss is questionable.
- Complexity issues:
   This solution introduce considerable complexity which might not be effective on large scale graphs. The authors fail to provide the complexity analysis and the trainning/execution time of the proposed solution.  The Top-K gating per node incurs extra computation and may be hard to scale to very large graphs or high expert counts without further efficiency tricks.
● Limited Task Scope
    Evaluation focuses solely on semi-supervised node classification. It remains unclear how MvCGE would perform on regression, link prediction, or graph-level tasks.
● Lack of reproducity, Source code and Settings are not provided
     This paper provide no source code with very complex design. The detailed settings are also missing, e.g. how to generate multi-view for single-view graphs , how to define the feature similarity threshold for connections? Settings on designs of differnt experts, number of experts, topK are not mentioned.

---

> ### Author Rebuttal · Authors · 2025-07-31
>
> We appreciate your constructive feedback and your recognition of our work. Many of your comments provide valuable insights that deserve further exploration. In the following, we systematically address each weakness (W) and question (Q) raised, and we would like to clarify several possible and potential misunderstandings.
>
> **W1. Confusion about $\mathcal{L}\_{\mathrm{dsp}}$**
>
> **Response:** We regret any confusion that may have occurred and would like to provide some clarifications. As mentioned at lines 239-240, the Maximum Mean Discrepancy (MMD) $\mathcal{M}^2_{\mathcal{H}}(\mathbb{P}\_{\mathcal{Q}}, \mathbb{P}\_{\mathcal{Q}'}) $ [1] is adopted to quantify the **discrepancy** between different graph experts. Then Eqn. (18) gives the MMD-based loss function:
> $$
> \mathcal{L}\_{\mathrm{dsp}} = -\frac{1}{L}\sum\_l \frac{1}{M(M-1)} \sum\_{m<m'}\mathcal{M}\^2\_{\mathcal{H}}\Big(\mathbb{P}\_{\mathcal{Q}^{(l)}\_m}, \mathbb{P}\_{\mathcal{Q}^{(l)}\_{m'}}\Big).
> $$
> Therefore, minimizing $ \mathcal{L}\_{\mathrm{dsp}} $ ***maximizes*** the differences among graph filters, which is ***consistent with our original design*** that promotes MvCGE to capture complementary information. We will add explicit statements like "minimizing the total loss maximizes MMD, thereby increasing inter-expert diversity" to avoid potential misunderstandings.
>
>
> **W2/Q1. Complexity concerns**
>
> **Response:** Our solution introduces a minor additional overhead but ***does not lead to a considerable increase in overall complexity*** [2, 3]. MoE has been widely adopted in LLMs precisely because it allows for the expansion of the model capacity while maintaining inference efficiency through sparse expert activates [4, 5]. In this work, we provide a detailed theoretical complexity analysis in ***Appendix B***. When employing $ E $ experts, the computational complexity of the gating mechanism within a single MvCGE layer is $ \mathcal{O}(NDE+|\mathcal{E}|D) $, while that of each hierarchical graph expert (e.g., GCN) is $ \mathcal{O}(ND^2+|\mathcal{E}|D) $. In practice, $ E $ is typically not much larger than the $ D $, so it will not be a bottleneck. The most time-consuming operation remains the graph convolution ($ \mathcal{O}(|\mathcal{E}|D) $). It is the gating mechanism that allows each sample to activate only a small subset of $K$ experts, even with dozens of graph experts. In addition, for spectral GNNs such as GCN, we can decouple the filter computation from the learnable parameters [6, 7], substantially reducing the number of filtering operations required. Empirically, Appendix A reports the results of MvCGE on two large-scale graphs, and here we further provide efficiency comparisons on three datasets to offer a clearer perspective:
>
> | Method | ACM (Time) | ACM (Memory) | Citeseer (Time) | Citeseer (Memory) | Flickr (Time) | Flickr (Memory) | UAI (Time) | UAI (Memory) |
> | --- | --- | --- | --- | --- | --- | --- | --- | --- |
> | **GCN** | 1.82s | 24.49 MB | 1.82s | 51.04 MB | 3.01s | 369.76 MB | 1.96s | 65.28 MB |
> | **MvCGE (3 experts, top 1)** | 3.60s | 25.41 MB | 3.33s | 52.21 MB | 4.91s | 374.16 MB | 3.52s | 67.00 MB |
> | **MvCGE (6 experts, top 2)** | 4.95s | 25.57 MB | 5.07s | 52.39 MB | 7.03s | 374.54 MB | 5.46s | 67.20 MB |
>
> We evaluated MvCGE under two settings: (1) 3 experts with each sample selecting the top-1 expert, and (2) 6 experts with each sample selecting the top-2 experts. The experiments were conducted on single-view graphs. Note that GCN processes only a single view, whereas MvCGE processes two views simultaneously. Nevertheless, MvCGE did not introduce significant computational overhead, which is consistent with our theoretical analysis and other empirical results.
>
> **W3/Q6. Generalization to other tasks**
>
> **Response:** In this work, we primarily validate the effectiveness of our model on the common graph learning task, semi-supervised node classification, rather than positioning MvCGE as a foundation model. As we demonstrated, it is straightforward to adapt the model to some graph tasks by simply replacing $ \mathcal{L}\_{\text{task}} $. We have conducted additional experiments on link prediction and graph-level classification, as shown below:
>
> | Method | IMDB-B | MUTAG | PTC |
> | --- | --- | --- | --- |
> | PATCHYSAN | 71.0 ± 2.2 | 92.6 ± 4.2 | 60.0 ± 4.8 |
> | DGCNN | 70 | 85.8 | 58.6 |
> | AWL | 74.5 ± 5.9 | 87.9 ± 9.8 | - |
> | MLP | 73.7 ± 3.7 | 84.0 ± 6.1 | 66.6 ± 6.9 |
> | GIN | 75.1 ± 5.1 | 89.4 ± 5.6 | 64.6 ± 7.0 |
> | GCN | 74.0 ± 3.4 | 85.6 ± 5.8 | 64.2 ± 4.3 |
> | GraphSAGE | 72.3 ± 5.3 | - | 63.9 ± 7.7 |
> | Ours | 76.5 ± 2.4 | 91.2 ± 3.9 | 68.6 ± 3.4 |
>
>
> | Category | Models | Cora MRR | Cora AUC |
> | --- | --- | --- | --- |
> | GNN | GCN | 32.5 ± 6.9 | 95.0 ± 0.3 |
> |   | GAT | 31.9 ± 6.1 | 93.9 ± 0.3 |
> |   | SAGE | 37.8 ± 7.8 | 95.6 ± 0.3 |
> |   | GAE | 30.0 ± 3.2 | 95.1 ± 0.3 |
> | Pairwise Info | SEAL | 26.7 ± 5.9 | 90.6 ± 0.8 |
> |    | BUDDY | 26.4 ± 4.4 | 95.1 ± 0.4 |
> |    | Neo-GNN | 22.7 ± 2.6 | 93.7 ± 0.4 |
> |    | NCN | 32.9 ± 6.3 | 96.8 ± 0.2 |
> |    | NCNC | 29.8 ± 5.4 | 96.7 ± 0.3 |
> |    | NBFNet | 37.7 ± 4.0 | 96.9 ± 0.3 |
> |    | PEG | 22.8 ± 1.8 | 94.5 ± 0.4 |
> | Ours | MvCGE | 36.3 ± 1.6 | 96.6 ± 0.3 |
>
> For the graph-level tasks, we followed the experimental settings of [8], while for the link prediction experiments, we adopted the settings from [9]. As can be seen, MvCGE continues to demonstrate strong performance on these tasks. We will consider evaluating the model in even more scenarios in future work. Thank you for your valuable suggestion.
>
>
>
> **W4/Q5. Reproducibility**
>
> **Response:** We clarify that we have included the anonymous repository link and hardware details in ***Appendix A.2***, which ensures reproducibility of our work. Here we further provide the detailed settings to address your concern:
> - For single-view graphs, we generate an additional feature-similarity $ k $NN graph ($ k=10 $, Euclidean distance) as the second view.
> - For the number of experts/Top-K, we fix them as $E=6$ and $K=2$ for Table 1 and 2, and analyze the impact of their choice in our paper (refer to Fig. 6), and we will add clear statements of the exact $ k $NN threshold and other tuning grids to an Implementation Details subsection.
> - Regarding the sensitivity for $ \alpha $ and $ \beta $, we have analyzed in the main body of our submission (refer to Fig. 5). As can be observed, performance is stable in $ \pm1 $ order-of-magnitude around the chosen $ \alpha $ and $ \beta $. We will revise our paper to include more detailed settings for reproducibility.
>
> **Q2. View definition & expert design**
>
> **Response:** For real multi-view datasets we use the provided edge sets. For single-view datasets we add a feature-similarity kNN graph (see above). All experts share the same base spectral GNN layer (e.g., GCN or ChebNet ). Diversity is imposed mainly by the mechanism of MoE and our proposed loss $\mathcal{L}\_{\text{dsp}}$.
>
>
> **Q3. Expert specialisation**
>
> **Response:** After training, it is indeed observed that different experts tend to focus on samples from a specific view. This forms the unique inductive bias of MvCGE, enabling the preservation of view-specific information. Regarding the substructures you have mentioned, since MvCGE is built upon graph spectral theory, different experts are able to learn distinct filter parameters, thereby capturing homophilic/heterophilic patterns. We have conducted visualization experiments on expert specialization and routing. Due to the NeurIPS 2025 rebuttal policy, we will include detailed discussion and related results in future versions.
>
> **Q4. Robustness to noise**
>
> **Response:** Owing to the presence of the multi-view aggregation module, MvCGE is theoretically capable of assigning low weights or even blocking those views that result in poor training loss (when using gating-based aggregation). As a result, the framework exhibits a certain degree of tolerance to noise. We conduct an experiment on ACM for a better understanding:
> | # Noisy Edges    | MaF1              | MiF1              |
> |-------------|-----------------|------------------|
> | 0.01\|𝓔\|   | 86.6 ± 0.4      | 86.6 ± 0.4      |
> | 0.05\|𝓔\|   | 85.2 ± 0.8      | 85.2 ± 0.8      |
> | 0.1\|𝓔\|    | 83.5 ± 0.5      | 83.5 ± 0.5      |
> | 0.5\|𝓔\|    | 81.7 ± 3.2      | 81.8 ± 3.2      |
> | \|𝓔\|       | 79.4 ± 5.4      | 79.7 ± 5.4      |
>
> Here, we construct a noisy view by randomly generating an adjacency matrix and test MvCGE's performance under different noise rates. We will include additional experiments on robustness to further discuss this aspect.
>
> ```
> [1] A kernel two-sample test. JMLR 2012.
> [2] Towards understanding the mixture-of-experts layer in deep learning. NeurIPS 2023.
> [3] Outrageously large neural networks: The sparsely-gated mixture-of-experts layer. ICLR 2017.
> [4] Switch transformers: Scaling to trillion parameter models with simple and efficient sparsity. JMLR 2022.
> [5] Scaling vision with sparse mixture of experts. NeurIPS 2021.
> [6] Semi-Supervised Classification with Graph Convolutional Networks. ICLR 2017.
> [7] Convolutional neural networks on graphs with chebyshev approximation, revisited. NeurIPS 2022.
> [8] How powerful are graph neural networks? ICLR 2018.
> [9] Evaluating graph neural networks for link prediction: Current pitfalls and new benchmarking. NeurIPS 2023.
> ```

---

> ### Author Response · Authors · 2025-08-08
>
> Thank you very much for your valuable suggestions. We will carefully revise our manuscript according to your comments and are ready to respond to any further questions you may have. We sincerely appreciate your time.

---

### Decision · Program_Chairs · 2025-09-17

**Decision:**

Accept (poster)

**Comment:**

This paper proposes a novel and well-motivated mixture-of-experts framework for multi-view GNNs, with clear motivation and strong empirical validation. Concerns regarding scalability, notation clarity, and theoretical depth were raised by reviewers, but the authors’ rebuttal addressed these points. All reviewers expressed satisfaction with the clarifications and advocated for acceptance. In summary, I agree with the reviewers and recommend accepting this paper.